# Evaluation of the Weed Infestation, Grain Health, and Productivity Parameters of Two Spelt Wheat Cultivars Depending on Crop Protection Intensification and Seeding Densities

**Małgorzata Haliniarz** [1] , **Dorota Gawęda** [1,*] , **Bożena Nowakowicz-Dębek** [2] ,
**Agnieszka Najda** [3] , **Sylwia Chojnacka** [1], **Justyna Łukasz** [1], **Łukasz Wlazło** [2] and
**Monika Różańska-Boczula** [4]

[1]  Department of Herbology and Plant Cultivation Techniques, Faculty of Agrobioengineering, University of Life Sciences in Lublin, Akademicka 13, 20-950 Lublin, Poland; malgorzata.haliniarz@up.lublin.pl (M.H.); schojnacka2@gmail.com (S.C.); justynalukasz1992@gmail.com (J.Ł.)

[2]  Department of Animal Hygiene and Environmental Hazards, Faculty of Animal Sciences and Bioeconomy, University of Life Sciences in Lublin, Akademicka 13, 20-950 Lublin, Poland; bozena.nowakowicz@up.lublin.pl (B.N.-D.); lukasz.wlazlo@up.lublin.pl (Ł.W.)

[3]  Department of Vegetable Crops and Medicinal Plants, Faculty of Horticulture and Landscape Architecture, University of Life Sciences in Lublin, Akademicka 15, 20-950 Lublin, Poland; agnieszka.najda@up.lublin.pl

[4]  Department of Applied Mathematics and Computer Science, Faculty of Production Engineering, University of Life Sciences in Lublin, Głęboka 28, 20-612 Lublin, Poland; monika.boczula@up.lublin.pl

*  Correspondence: dorota.gaweda@up.lublin.pl; Tel.: +48-81-445-67-69

**Abstract:** Spelt wheat is one of the oldest wheat with very high nutritional value. It does not have particular climatic requirements and tolerates adverse environmental conditions well. The versatile advantages of spelt wheat make it attractive to farmers, plant breeders, food technologists, and consumers. The aim of this study was to determine the effect of different crop protection systems and seeding densities on yield, weed infestation, and grain health of the spelt wheat cultivars "Rokosz" and "Schwabenspelz". The research showed that the spelt wheat cultivars studied responded differently to production intensification. The use of crop protection chemicals in the crop of the cultivar "Rokosz" resulted in lower weed infestation and in obtaining higher yields. In the case of the cultivar "Schwabenspelz", production intensification did not have a significant effect on its productivity and quantitative weed infestation parameters. Therefore, this cultivar can be recommended for cultivation in farms with extensive farming methods, for example, in organic farms. In both cultivars studied, an increase in seeding density and chemical plant protection with fungicide caused lower grain contamination with mycotoxins, and the content of individual mycotoxins did not exceed the maximum levels set for grain intended for food and animal feed purposes.

**Keywords:** spelt wheat; cultivars; crop protection methods; seeding rate; weed infestation; yield; fungi; mycotoxins

## 1. Introduction

Spelt wheat (*Triticum aestivum* ssp. *spelta*) probably originates from the region of southeastern Asia and is one of the oldest wheat subspecies [1,2]. The beginnings of its cultivation date back to the Stone Age. Archaeological investigations reveal that, starting with the Bronze Age, spelt was the most important consumption crop in Europe for many years [3,4]. However, its small production

potential and difficulties with grain dehulling became the reason why the cultivation of this crop was abandoned in favor of more productive common wheat varieties [4,5].

Changing trends both in cultivation of agricultural crops and in human nutrition have been observed in recent years. People have started to attach a great importance to increased biological diversity of agricultural plant communities, reduced use of industrial production means in agriculture, and consumption of products with health promoting properties [3,4,6]. During the last 20 years, due to its nutritional qualities, spelt wheat was again introduced into cultivation, primarily through the organic farming system [7–11]. In Poland, similar to other European countries, the cultivation of spelt wheat is concentrated in organic farms and the interest in this cereal is continually growing [12]. The estimated cropped area of spelt in 2015 was about 225 ha [13]. The productivity of this species in Poland varies greatly. Grain yield can exceed 6 t ha$^{-1}$ [14], while under conditions unfavorable for growing common wheat, yields can be even higher than those obtained for common wheat [12]. In comparison with common wheat, spelt grain contains more protein, gluten, and fat, but less dietary fiber. It is a rich source of silicon, minerals, and group B vitamins. Moreover, it contains substances with antioxidant properties [5,15–18]. Spelt wheat does not have high climatic and soil requirements, and tolerates adverse environmental conditions well [19–22]. It is characterized by a rather high level of resistance to ear and grain diseases caused by fungal pathogens [8,23] and, due to its high tillering ability, competes with weeds better than common wheat [24]. These characteristics allow this crop to be grown without excessive use of crop protection products and mineral fertilizers, and hence it is considered to be environmentally friendly [25]. Studies conducted by different researchers have shown great variations in productivity and quality parameters between different spelt wheat varieties [18,21,26,27], as well as their varying response to soil and climatic conditions [22] and agronomic factors [18,27–29]. Most research is focused on organic cultivation of spelt [17,21,22,30], but there is little information how spelt wheat responds to production intensification. According to Rachoń et al. [31] and Andruszczak et al. [32], higher grain yields were obtained under full chemical protection conditions compared to extensive cultivation conditions. Therefore, a study was undertaken to explain how crop protection intensification and different seeding densities affect yield, yield components, weed infestation, and grain health of two winter wheat spelt cultivars. Taking into account the genetic characteristics of spelt wheat, the experiment assumed that (i) the absence of any crop protection or mechanical protection and application of herbicide alone would not contribute to a significant decrease in grain yield and health, compared to full chemical protection; (ii) pro-ecological technology involving the complete abandonment of crop protection or the use of only harrowing would not deteriorate the productivity of spelt wheat and its health, in comparison with chemical protection; (iii) in treatments with a higher seeding rate, the denser spelt crop would be more competitive against weeds, which would contribute to reduced weed infestation and would not increase grain infection with fungal pathogens, in consequence allowing a satisfactory yield to be obtained; and (iv) the response of spelt wheat to production technologies is cultivar-dependent.

## 2. Materials and Methods

### 2.1. Location of the Experiment—Soil and Climatic Conditions

The field study was carried out for three growing seasons, 2012/2013, 2013/2014, and 2014/2015, at the Czesławice Experimental Farm (51°18′23′′ N, 22°16′2′′ E), belonging to the Lublin University of Life Sciences, Poland. The experiment on growing spelt wheat was established on a loess-derived Luvisol. The arable layer of the soil was characterized by high availability of phosphorus (P, 76.3–77.7 mg kg$^{-1}$ soil) and potassium (K, 117.8–132.3 mg kg$^{-1}$ soil), as well as medium availability of magnesium (Mg, 79–85 mg kg$^{-1}$ soil), slightly acidic pH (in 1 M KCl, 6.1–6.4), and a low humus content of 1.59–1.63%.

The 2013/2014 season could be described as very wet since the total rainfall was higher by 281 mm than in the 2014/2015 season, 147.1 mm higher than in 2012/2013, and 206.3 mm higher than the

1963–2010 average (Table 1). In the years 2013 and 2014, the amount of rainfall in May and June much exceeded the long-term average (LTA). In August 2014, there was 32.5 mm more rain than the LTA. In 2015 the amount of rainfall in May was high, but June and August were very dry with only 13.5 and 5.9 mm of rain, respectively. The temperature ranges in the 2013/2014 and 2014/2015 growing seasons were similar. The mean temperature in these years was higher by 0.9 °C in the 2013/2014 season and by 1 °C in 2014/2015. In the 2012/2013 season, the mean temperature was the same as the long-term average. In all the growing seasons, July was warmer than over the period 1963–2010.

**Table 1.** The rainfall and mean monthly air temperature in the growing season of spelt wheat, recorded by the Meteorological Station in Czesławice (Poland). LTA is the long term average.

| Years | Months | | | | | | | | | | | | September–August |
|---|---|---|---|---|---|---|---|---|---|---|---|---|---|
| | September | October | November | December | January | February | March | April | May | June | July | August | |
| | **Rainfalls (mm)** | | | | | | | | | | | | Sum |
| 2012/ 2013 | 40.4 | 110.7 | 29 | 17.4 | 60.3 | 30 | 37.6 | 53.2 | 103.3 | 108.3 | 44.3 | 26.6 | 661.1 |
| 2013/ 2014 | 49.5 | 7.3 | 60.6 | 13.7 | 54.5 | 5.8 | 49.1 | 63.9 | 230.2 | 110.2 | 61.4 | 102 | 808.2 |
| 2014/ 2015 | 21.8 | 27.5 | 24.1 | 57.8 | 50.9 | 15.8 | 48.6 | 39.1 | 169.6 | 13.5 | 52.6 | 5.9 | 527.2 |
| LTA 1963–2010 | 59.5 | 45.6 | 41 | 36.9 | 30.3 | 29.2 | 31.3 | 42.4 | 63.5 | 72.7 | 80 | 69.5 | 601.9 |
| | **Temperature (°C)** | | | | | | | | | | | | Mean |
| 2012/ 2013 | 14.6 | 7.7 | 5 | −3.4 | −4.4 | −1.3 | −2.6 | 7.4 | 14.9 | 18.1 | 18.7 | 18.7 | 7.8 |
| 2013/ 2014 | 11.3 | 9.5 | 4.9 | 1.7 | −2.9 | 0.3 | 4.9 | 8.9 | 13 | 15.2 | 19.6 | 18.3 | 8.7 |
| 2014/ 2015 | 14.0 | 9.7 | 4.6 | −0.1 | 1.0 | −1.1 | 2.8 | 6.5 | 11.5 | 16.1 | 19 | 21.9 | 8.8 |
| LTA 1963–2010 | 13.1 | 7.9 | 2.9 | −1.3 | −3.0 | −1.7 | 1.8 | 7.7 | 13.6 | 16.5 | 18.3 | 17.7 | 7.8 |

*2.2. Experimental Design and Agronomic Practices*

A field experiment was set up using a split-block design in three replicates. Two winter spelt wheat cultivars, the German cultivar "Schwabenspelz" and the Polish cultivar "Rokosz", were evaluated in this study. The experiment included the following factors:

I—crop protection

1. pro-ecological:

A—untreated control plots (no herbicide, fungicide, or insecticide treatment);
B—harrowing in early spring at the beginning of the growing season;

2. chemical:

C—application of the herbicides Sekator 125 OD (Bayer AG, Leverkusen, Germany, active ingredients (a.i.) amidosulfuron, iodosulfuron, mefenpyr-diethyl) at a rate of 150 mL ha$^{-1}$ and Attribut 70 WG (Bayer AG, Leverkusen, Germany; a.i. propoxycarbazone-sodium) at a rate of 60 g ha$^{-1}$ at BBCH (**B**iologische **B**undesanstalt, **B**undessortenamt und **CH**emische Industrie) stages 22–24 of spelt wheat;

D—application of complete chemical protection—the herbicides Sekator 125 OD at a rate of 150 mL ha$^{-1}$ and Attribut 70 WG at a rate of 60 g ha$^{-1}$ at BBCH 22–24 of spelt wheat; the growth

retardant Cerone 480 SL (Bayer AG, Leverkusen, Germany, a.i. ethephon) at BBCH 30–31 at a rate of 0.75 L ha$^{-1}$, the fungicide Wirtuoz 520 EC (Corteva, Inc., Wilmington, DE, USA, a.i. prochloraz, tebuconazole, proquinazid) at BBCH 24–25 and 32–33 at a rate of 1 L ha$^{-1}$, as well as the insecticide Decis 2,5 EC (Bayer AG, Leverkusen, Germany, a.i. deltamethrin) at BBCH 37–39 at an amount of 0.25 L ha$^{-1}$.

II—seeding density:

1. optimum—130 kg ha$^{-1}$ of cv. "Rokosz" seeds and 200 kg ha$^{-1}$ of cv. "Schwabenspelz" spikelets;
2. increased—200 kg ha$^{-1}$ of cv. "Rokosz" seeds and 350 kg ha$^{-1}$ of cv. "Schwabenspelz" spikelets.

Mineral fertilization was applied at the following rates: N, 50 kg ha$^{-1}$; P, 26.2 kg ha$^{-1}$; and K, 58.1 kg ha$^{-1}$. All phosphorus and potassium fertilizers and part of the nitrogen fertilizers (20 kg N) were applied before sowing the spelt wheat. The remaining portion of nitrogen fertilizers was applied right after the beginning of the growing season. The sown and harvested plot area was 13.5 m$^2$, per repetition. Winter wheat was the previous crop for spelt wheat. Tillage was typical for common wheat cultivation. The spelt wheat was sown in the third decade of September 2012–2014 and harvested in the second decade of August 2013–2015.

### 2.3. Evaluation of Yield and Weed Infestation

Approximately 3–4 weeks after herbicide treatment and at the dough stage of spelt wheat (BBCH 83–85), evaluation of weed infestation of the spelt crop was made using the dry-weight-rank system. The evaluation involved determination of the botanical composition of weeds, as well as their number and dry weight per 1 m$^2$. The sampling area was delineated with a 0.5 × 1 m quadrat frame in two randomly selected places in each plot. Weed nomenclature followed Mirek et al. [33].

Plant height and ear length were determined based on 30 randomly selected plants from each plot. Weight of grains per ear was determined based on the average results for 30 randomly selected ears per plot. Ear density was calculated in two randomly chosen places with an area of 0.5 m$^2$ per plot. Yield was estimated based on grain harvested from the individual plots and expressed on a per hectare basis.

### 2.4. Determination of Fungal Contamination of Grain

Samples of about 20 g of spelt grain (from each plot) were homogenized by hand for 5 min and then sedimented for 15 minutes in sterile dilution liquid with approximately 100 μL of Tween 80 (Sigma-Aldrich, St. Louis, MO, USA). The dilutions were prepared and inoculated on sterile Sabouraud Chloramphenicol Agar (BTL Poland Ltd., Warsaw, Poland). To determine the total number of fungi, the plates were incubated at a temperature of 25 ± 0.2 °C for 7 days according to the relevant standard [34,35]. The colonies were transferred to microcultures and species were determined using Watanabe's key [36]. Each sample was analyzed in triplicate and the results presented as the number of fungi per 1 g of the tested material (cfu g$^{-1}$).

### 2.5. Determination of Mycotoxin Contamination of Grain

The content of deoxynivalenol (DON) and its derivatives: 3-O-acetylDON (3AcDON), 15-O-acetylDON (15AcDON), and zearalenone (UAE) were determined according to the authors' method developed in the Laboratory of Vegetable and Herb Raw Material Quality Research: "Determination of deoxynivalenol and its derivatives, nivalenol and zearalenone by HPLC method with purification by immunological affinity column in plant raw materials". The presence of trichothecene mycotoxins such as T-2 toxin (T-2) and its derivatives HT-2 toxin (HT-2), nivalenol (NIV), DAS (diacetoxyscirpenol), and fuzarenone X were analyzed by gas chromatography following the procedure given by Valle-Algarra et al. [37].

### 2.5.1. Sample Preparation and Extraction

A 5 g sample of spelt grain previously finely ground with a laboratory mill (FW135, ChemLand, Stargard Szczeciński, Poland) was placed in a test tube and treated with 20 mL of acetonitrile (ACN): H2O so lution (85:15). The diluted samples were shaken with a centrifuge (30 min, 21.000× g, Universal 32, Hettich, Tuttlingen, Germany) and left for 6 h, after which time it was again shaken for 30 min. The extract was filtered through a 0.2 Pm membrane syringe filter (Costar X, Corning Inc., Salt Lake City, UT, USA). Then, 5 mL of filtrate was collected for thorough purification and applied to a MycoSep 225 Trich column (Romer Laboratories, Union, MO, USA). Two mL of purified eluents were evaporated to dryness at 50 °C, and the residue dissolved in 0.5 mL of mobile phase (10% aqueous acetonitrile) after which the mycotoxin content was determined.

### 2.5.2. HPLC Analysis Conditions

A LaChrom–Merck HPLC liquid chromatograph (Merck, Darmstadt, Germany) was equipped with a DAD diode detector (L-7450), binary pump (L-7100), degasser (L-7612), thermostat (L-7360), Rheodyne injector, and steel column LiChrospher 100 RP C18 (250 × 4 mm) filled with stationary phase (particle diameter—dp = 5 μM). UV detection was performed at 220 nm (deoxynivalenol, 3-O-acetylDON, 15-O-acetylDON) and 220–440 (zearalenone) at 28 °C and a flow rate of 0.800 mL min$^{-1}$. The mobile phase was aqueous acetonitrile. Separation was achieved in an isocratic water: acetonitrile (8:92) analysis by deoxynivalenol, 3-O-acetylDON, 15-O-acetylDON, and (63:37) by analysis of zearalenone. To increase the separation, 1% acetic acid was added to the mobile phase. Identification of analyzed compounds was made on the basis of retention time. Quantitative assessment was carried out using the external standard method, after creating a calibration curve using pure reference substances.

### 2.5.3. GC/MS Analysis Conditions

Separation of trichothecenes, nivalenol, and fuzarenone X was carried out using a Varian CP-3800 gas chromatograph (Varian Medical Systems, Palo Alto, CA, USA) equipped with an detector ECD (instant connect Electron Capture Detector) and columns VF-1ms 30 m × 0.32 mm ID (Column Internal Diameter), DF (film thickness) = 0.25 μm from Varian. The temperature of the dispenser chamber was 250 °C and the detector 300 °C. The following oven temperature gradient was used: 80 °C for 1 min, increment 40 °C for 1 min to 160 °C, 3 °C for 1 min to 178 °C, 2 °C for 1 min to 240 °C, 40 °C for 1 min to 270 °C, 270 °C for 1 min. Nitrogen was used as carrier gas at a flow rate of 1 mL for min.

### 2.5.4. Chemical Reagents

Organic solvents with spectroscopic purity were obtained from Merck (Darmstadt, Germany). All deoxynivalenol (DON) purity ≥ 95% (reference standard), 3-O-acetylDON (3AcDON) purity ≥ 95% (reference standard), 15-O-acetylDON (15AcDON) purity ≥ 95% (reference standard), zearalenone (UAE) purity ≥ 95% (reference standard), T-2 toxin (T-2) purity ≥ 95% (reference standard) and its derivatives HT-2 toxin (HT-2) purity ≥ 95% (reference standard), nivalenol (NIV) purity ≥ 95% (reference standard), DAS—diacetoxyscirpenol purity ≥ 95% (reference standard) and fuzarenone X purity ≥ 95% (reference standard) were from Sigma-Aldrich (St. Louis, MO, USA).

### 2.6. Statistical Analysis

The results were statistically analyzed using analysis of variance (Statistica PL 13.3, StatSoft Polska, Krakow, Poland). The distribution conformity with normal distribution was verified with the Shapiro–Wilk test, while homogeneity of the variance was tested with Levene's test. In the case of any feature where a significant interaction was found between the factor and year, interaction effects were analyzed separately for each year of the study. If no significant interactions were found between the factors, then Tukey's multiple comparison test was used to compare differences between the means for main factors (crop protection, seeding density, cultivar). Confidence half-intervals were

calculated by Tukey's test at a significance level of 0.05. Significant differences between the two groups of treatments, pro-ecological protection (treatments A and B) and chemical protection (treatments C and D), were evaluated based on the Scheffe test ($\alpha$ = 0.05). Total number of fungi in grain is presented in log CFU g$^{-1}$.

## 3. Results

### 3.1. Weed Infestation of Spelt Wheat

The statistical analysis of the weed infestation parameters for the crop of the "Rokosz" spelt wheat cultivar, as evaluated 3–4 weeks after herbicide application, revealed significant differences in the traits between years (Table 2). The highest number of weeds was found in the first year of the study in the treatments where no herbicide was used (treatments A and B) and in the second year of the study in the control treatment. Weeds produced the highest biomass in 2013 in treatments A and B. Harrowing done as the only weed control operation (treatment B) proved to be ineffective in reducing weed infestation because the number and dry weight of weeds were similar to those found in the control treatment (A) in each year of the study. However, it was only in 2013 that weed infestation in this treatment was significantly higher than in the treatments where the herbicide was applied. In the years 2014 and 2015, as well as in treatments C and D in 2013, weed infestation of the crop was at a similar level. At the dough stage of wheat, crop protection method and year were not found to significantly affect number of weeds. The highest weight of weeds was determined in the first year of the study in the treatments without herbicide application (A and B). In the other treatments, the weight of weeds was similar. The results for the three-year study showed that in cv. "Rokosz", herbicide application had a significant impact on weed infestation of the crop. At the first evaluation of weed infestation, the average number of weeds in the chemical protection treatments (C and D) was more than one-half lower, while their dry weight was four times lower than in the pro-ecological treatments A and B. At the dough stage of cv. "Rokosz", similar relationships were observed as those made 3–4 weeks after herbicide treatment.

**Table 2.** The weed infestation indicators in the spelt wheat cv. "Rokosz" showing the interaction between crop protection and years of research, and comparison of the pro-ecological and chemical crop protection. NW I—number of weeds 3–4 weeks after herbicide treatment, WW I—air-dry weight of weeds 3–4 weeks after herbicide treatment, NW II—number of weeds at the dough stage of spelt wheat (BBCH 83–85), WW II—air-dry weight of weeds at the dough stage of spelt wheat (BBCH 83–85). Different letters denote significant differences ($p \leq 0.05$). The same letter means not significantly different values.

| Years | Crop Protection | NW I (no. m$^{-2}$) | WW I (g m$^{-2}$) | NW II (no. m$^{-2}$) | WW II (g m$^{-2}$) |
|---|---|---|---|---|---|
| 2013 | A | 90.8 a | 68.66 a | 86.3 a | 158.06 a |
| | B | 78.7 ab | 62.12 a | 58.3 a | 119.1 ab |
| | C | 27.2 c | 5.68 b | 18 a | 21.7 c |
| | D | 28.5 c | 8.18 b | 21.2 a | 25.33 c |
| 2014 | A | 53 abc | 25.4 b | 56 a | 39.39 bc |
| | B | 37.7 bc | 13.71 b | 42 a | 19.37 c |
| | C | 28 c | 7.79 b | 32.3 a | 13.17 c |
| | D | 31.3 bc | 6.51 b | 42 a | 19.87 c |
| 2015 | A | 21.3c | 10.99b | 12 a | 13.13 c |
| | B | 22.7c | 13.6b | 13.3 a | 17.25 c |
| | C | 13.3c | 7.02b | 7.8 a | 5.09 c |
| | D | 13.3c | 8.31b | 8.7 a | 9.11 c |
| Comparison of the pro-ecological and chemical crop protection (mean for 2013–2015) | | | | | |
| pro-ecological (A, B) | | 50,7 a | 32.41 a | 44.7 a | 61.05 a |
| chemical (C, D) | | 23,6 b | 7.25 b | 21.7 b | 15.71 b |

In the cv. "Schwabenspelz" crop, the evaluation of weed infestation performed 3–4 weeks after herbicide treatment demonstrated that significantly the lowest number of weeds was found under chemical protection conditions (treatments C and D) (Table 3). Nonetheless, the number of weeds in the harrowing treatment and in the treatment where the herbicide was applied alone (treatment C) was similar. After application of chemical protection (treatments C and D), the dry weight of weeds was lower by 76.4% and 69.4% than in the control treatment (A). The statistical calculations showed that at the first evaluation of weed infestation, the average number and dry weight of weeds found in the chemical treatments (C and D) were significantly lower than those determined for the pro-ecological treatments (A and B) by 42.7% and 70.5%, respectively. At the dough stage of wheat, crop protection method was not shown to have an effect on the number and dry weight of weeds in the cv. "Schwabenspelz" crop.

**Table 3.** The impact of crop protection system on weed infestation indicators in the spelt wheat cv. "Schwabenspelz" (mean for 2013–2015). NW I—number of weeds 3–4 weeks after herbicide treatment, WW I—air-dry weight of weeds 3–4 weeks after herbicide treatment, NW II—number of weeds at the dough stage of spelt wheat (BBCH 83–85), WW II—air-dry weight of weeds at the dough stage of spelt wheat (BBCH 83–85). Different letters denote significant differences ($p \le 0.05$). The same letter means not significantly different values.

| Crop Protection | NW I (no. m$^{-2}$) | WW I (g m$^{-2}$) | NW II (no. m$^{-2}$) | WW II (g m$^{-2}$) |
|---|---|---|---|---|
| A | 44.3 a | 26.35 a | 33.7 a | 32.58 a |
| B | 43.8 ab | 21.95 a | 27.8 a | 25.22 a |
| C | 26.2 bc | 6.21 b | 16.8 a | 8.42 a |
| D | 24.2 c | 8.05 b | 22.9 a | 12.89 a |
| Comparison of the pro-ecological and chemical crop protection | | | | |
| pro-ecological (A, B) | 44 a | 24.15 a | 30.7 a | 28.9 a |
| chemical (C, D) | 25.2 b | 7.13 b | 19.9 a | 10.66 a |

On average over the period 2013–2015, seeding density was not found to have a significant influence on the number and dry weight of weeds in the cv. "Rokosz" the cv. "Schwabenspelz" crop at both times that weed infestation was evaluated (Figure 1). Nonetheless, it could be observed that a higher seeding density had a more beneficial effect on weed infestation, as demonstrated by the lower number and weight of weeds in the crop.

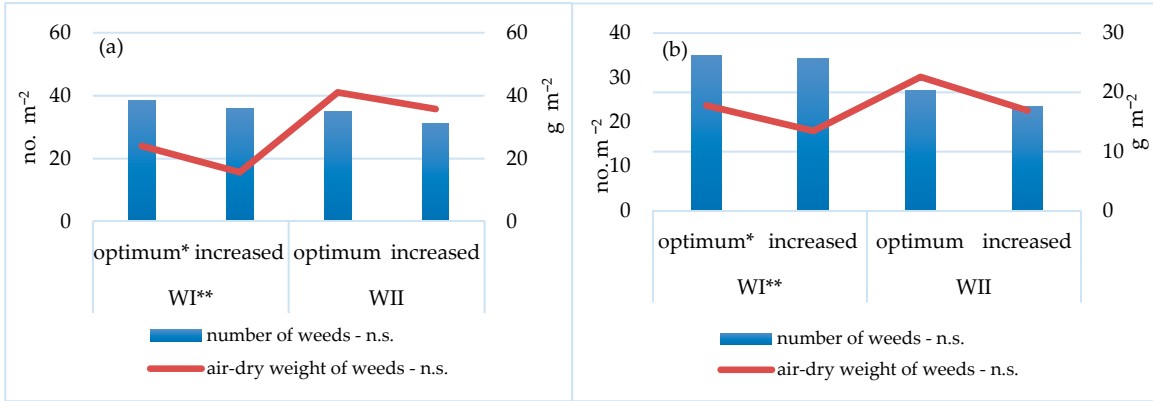

**Figure 1.** The impact of seeding density on weed infestation indicators in the spelt wheat cultivars "Rokosz" (**a**) and "Schwabenspelz" (**b**) (mean for 2013–2015). Note: * seeding density, **WI—number and air-dry weight of weeds 3–4 weeks after herbicide treatment, WII—number and air-dry weight of weeds at the dough stage of spelt wheat (BBCH 83–85), n.s.—not significant differences.

The analysis of the quantitative weed infestation parameters did not reveal significant differences in the number of weeds between the studied cultivars, but it proved a significantly lower biomass of weeds in the tall cultivar "Schwabenspelz". At the first evaluation of weed infestation, 3–4 weeks after herbicide treatment, the weed biomass in the crop of this cultivar was determined to be 15.64 g m$^{-2}$ and it was 21% lower than for cv. "Rokosz". During the evaluation of weed infestation at the dough stage of spelt wheat, the difference in the biomass of weeds occurring in the crops of the cultivars studied was even greater, since for cv. "Rokosz" it was almost twice higher than for cv. "Schwabenspelz". At the first evaluation of weed infestation, *Viola arvensis*, *Geranium pusillum*, and *Apera spica-venti* were predominant in the spelt wheat crop (Figure 2a,b). In both cultivars, chemical crop protection significantly reduced the number of *Apera spica-venti* and *Viola arvensis* (Figure 2a). Seeding density did not cause differences in numbers of the dominant weed species in the crop (Figure 2b). At the dough stage (BBCH 83–85) of spelt wheat, the number of *Viola arvensis* plants was similar among all the treatments, while the number of *Apera spica-venti* was significantly lower on chemical crop protection treatments, compared to treatments A and B. (Figure 3a). In the cv. "Schwabenspelz", an increased seeding density reduced significantly the numbers of *Apera spica-venti* (Figure 3b).

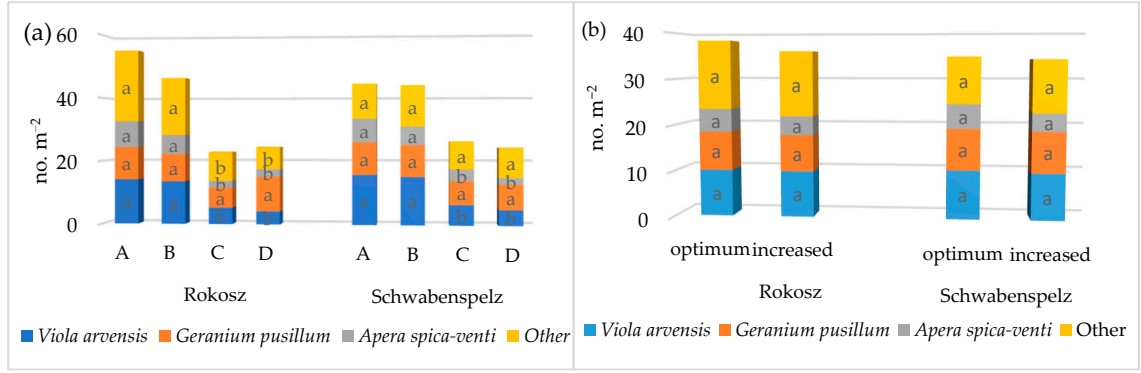

**Figure 2.** The impact of crop protection system on number of dominant weeds species 3–4 weeks after herbicide treatment in the spelt wheat cultivars "Rokosz" and "Schwabenspelz" (mean for 2013–2015). Different letters denote significant differences ($p \leq 0.05$) of number of weed species among different crop protection methods (A, B, C, D) (**a**) and seeding density (**b**). The same letter means not significantly different values.

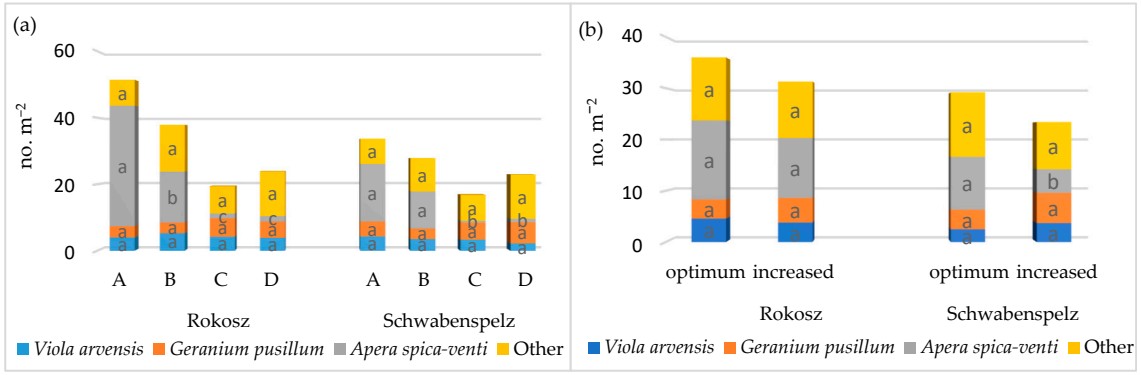

**Figure 3.** The impact of crop protection system on number of dominant weeds species at the dough stage (BBCH 83–85) in the spelt wheat cultivars "Rokosz" and "Schwabenspelz" (mean for 2013–2015). Different letters denote significant differences ($p \leq 0.05$) of number of weed species among different crop protection methods (A, B, C, D) (**a**) and seeding density (**b**). The same letter means not significantly different values.

### 3.2. Yield and Yield Components of Spelt Wheat

On average over the three-year study period, the crop protection system did not cause significant differences in the yield components of the cultivar "Rokosz" (Table 4). The comparison of pro-ecological crop protection systems (A, B) and chemical crop protection did not reveal any significant differences in the yield components.

**Table 4.** The impact of crop protection system on selected yield components of the spelt wheat cv. "Rokosz" (mean for 2013–2015). HP—height of plant, LE—length of ear, NE—number of ears per m$^2$, WG—weight of grains per ear. The same letter means not significantly different values.

| Crop Protection | HP (cm) | LE (cm) | NE (no. m$^{-2}$) | WG (g) |
|---|---|---|---|---|
| A | 106a | 8.3a | 417a | 1.1a |
| B | 104.2a | 8.4a | 421a | 1.15a |
| C | 105.5a | 8.6a | 413a | 1.15a |
| D | 99.9a | 8.5a | 459a | 1.33a |
| Comparison of the pro-ecological and chemical crop protection | | | | |
| pro-ecological (A, B) | 105.1a | 8.4a | 419a | 1.13a |
| chemical (C, D) | 102.7a | 8.5a | 436a | 1.24a |

The statistical analysis showed cv. "Rokosz" grain yield to be significantly dependent on crop protection system and year (Table 5). In the first year of the experiment, the highest grain yield was obtained from the treatment where full chemical crop protection was used (treatment D). In the years 2014 and 2015, crop protection intensification did not contribute to increased yields of cv. "Rokosz". In 2015, higher yields were obtained in all crop protection treatments than in 2014, while treatments A, B, and C were higher than in 2013. The statistical analysis, performed based on the means for the three-year study period, revealed that the average grain yield from the chemical protection treatments (C, D) was 19.1% higher than in the pro-ecological treatments (A, B).

**Table 5.** The grain yield of the spelt wheat cv. "Rokosz". The interaction between crop protection and years of research, and comparison of the pro-ecological and chemical crop protection. Different letters denote significant differences ($p \leq 0.05$).

| Crop Protection | 2013 | 2014 | 2015 |
|---|---|---|---|
| | | t ha$^{-1}$ | |
| A | 2.43 d | 3.23 cd | 5.62 ab |
| B | 2.73 d | 3.31 cd | 5.66 ab |
| C | 3.57 cd | 3.19 cd | 5.73 a |
| D | 4.7 abc | 4.05 bcd | 6.1 a |
| Comparison of the pro-ecological and chemical crop protection (mean for 2013–2015) | | | |
| pro-ecological (A, B) | | 3.83 a | |
| chemical (C, D) | | 4.56 b | |

As far as the analyzed yield components for cv. "Schwabenspelz" are concerned, the crop protection methods significantly modified plant height, but they had no effect on ear length, density, or grain weight per ear (Table 6). Wheat in the treatment that included only harrowing produced the highest plants (treatment B); the plant height was significantly lower in the treatment with full chemical protection (D). The different crop protection methods used did not affect yield of the spelt wheat cultivar "Schwabenspelz". The Scheffe test did not show significant differences in grain yield between pro-ecological technologies and chemical protection of the cv. "Schwabenspelz" crop.

**Table 6.** The impact of crop protection system on selected yield components and grain yield of the spelt wheat cv. "Schwabenspelz" (mean for 2013–2015). HP—height of plant, LE—length of ear, NE—number of ears per m$^2$, WG—weight of grains per ear, Y—yield for the crop protection methods A–D.

| Crop Protection | HP (cm) | LE (cm) | NE (no. m$^{-2}$) | WG (g) | Y (t ha$^{-1}$) |
|---|---|---|---|---|---|
| A | 127.9 ab | 11.5 a | 357 a | 1.03 a | 3.12 a |
| B | 131.4 a | 11.5 a | 353 a | 1.14 a | 3.02 a |
| C | 128.9 ab | 11.2 a | 372 a | 1.09 a | 3.16 a |
| D | 119.3 b | 11.4 a | 389 a | 1.13 a | 3.56 a |
| Comparison of the pro-ecological and chemical crop protection | | | | | |
| pro-ecological (A, B) | 129.6 a | 11.5 a | 355 a | 1.09 a | 3.07 a |
| chemical (C, D) | 124.1 a | 11.3 a | 381 a | 1.11 a | 3.36 a |

Different letters denote significant differences ($p \leq 0.05$). The same letter means not significantly different values.

On average over the period 2013–2015, seeding density significantly modified only the ear length of the wheat cultivar "Rokosz", but significantly the highest value of this parameter was achieved at an optimal seeding density (Table 7). The seeding density factor did not cause significant differences in the other biometric characteristics and yield of cv. "Rokosz". Also, seeding density was not found to have a significant effect on the yield components and grain yield of cv. "Schwabenspelz".

**Table 7.** The impact of seeding density on selected yield components and grain yield of the spelt wheat cultivars "Rokosz" and "Schwabenspelz" (mean for 2013–2015). HP–height of plant, LE–length of ear, NE–number of ears per m$^2$, WG–weight of grains per ear, Y–yield.

| Seeding Density | HP (cm) | LE (cm) | NE (no. m$^{-2}$) | WG (g) | Y (t ha$^{-1}$) |
|---|---|---|---|---|---|
| cv. "Rokosz" | | | | | |
| optimum | 105.7 a | 8.7 a | 412 a | 1.22 a | 4.25 a |
| increased | 102.2 a | 8.3 b | 443 a | 1.15 a | 4.14 a |
| cv. "Schwabenspelz" | | | | | |
| optimum | 125.9 a | 11.3 a | 350 a | 1.11 a | 3.24 a |
| increased | 127.9 a | 11.5 a | 386 a | 1.09 a | 3.25 a |

Different letters denote significant differences ($p \leq 0.05$). The same letter means not significantly different values.

Comparing the cultivars studied, it was found that the spelt wheat cultivar "Schwabenspelz" was characterized by significantly higher plants (by 22.7%) and longer ears (by 29.5%) relative to cv. "Rokosz". The comparison of the spelt wheat cultivars concerning the other biometric characteristics did not show significant differences. The grain yield of the spelt wheat cultivar "Rokosz" was significantly higher than that of cv. "Schwabenspelz", on average by 31.8%.

*3.3. Grain Health*

The results showed that the total number of fungi in cv. "Rokosz" grain ranged from 3.11 log CFU g$^{-1}$ in treatment D with a increased seeding density to 3.87 log CFU g$^{-1}$ in treatment A with an optimal seeding density, while in cv. "Schwabenspelz" grain ranged from 3.82 log CFU g$^{-1}$ in treatment D with a higher seeding density to 4.46 log CFU g$^{-1}$ in treatment B with an optimal seeding density (Figure 4a,b). In the grain with chemical crop protection (treatments C and D) in cv. "Rokosz" the total number of fungi was statistically significantly lower ($p \leq 0.05$). The fungi *Penicillium* sp., *Penicillium expansum*, *Candida* sp., *Aureobasidium* sp., *Aspergillus* sp., *Aspergillus flavus*, *Aspergillus terreus*, *Cladosporium* sp., *Cladosporium cladosporides*, *Ulocladium* sp., *Fusarium* sp., *Rhodotorula* sp., *Rhizopus* sp., *Trichophyton* sp., and *Phoma* sp. were found in different ratio in spelt wheat (Figures 5 and 6).

In cv. "Rokosz" grain, *Fusarium* sp. (19.4–76.7%) and *Aspergillus* sp. (11.9–38.9%) were the most abundant species, while in cv. "Schwabenspelz" grain the most abundant were *Cladosporium* sp. (7.23–53.3%) and *Fusarium* sp. (4.76–38.5%).

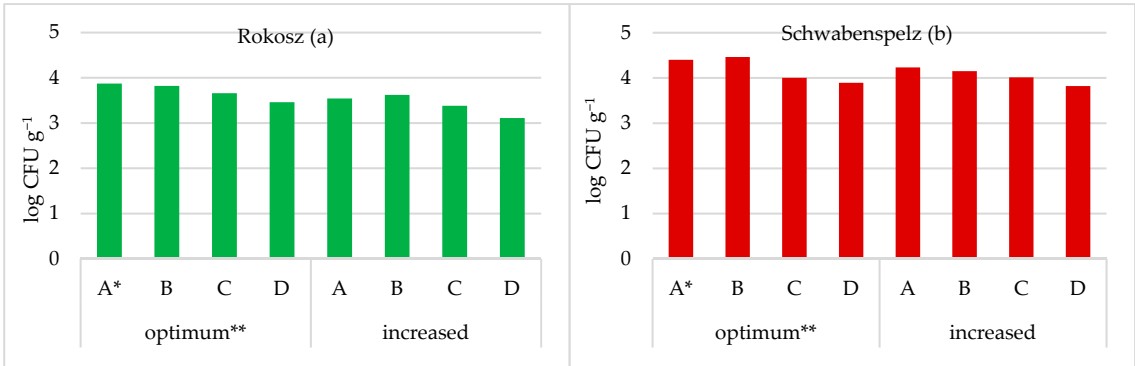

**Figure 4.** Total number of fungi in the grain of spelt wheat cv. "Rokosz" (**a**) and cv. "Schwabenspelz" (**b**) (mean for 2014–2015). Note: * crop protection methods, ** seeding density.

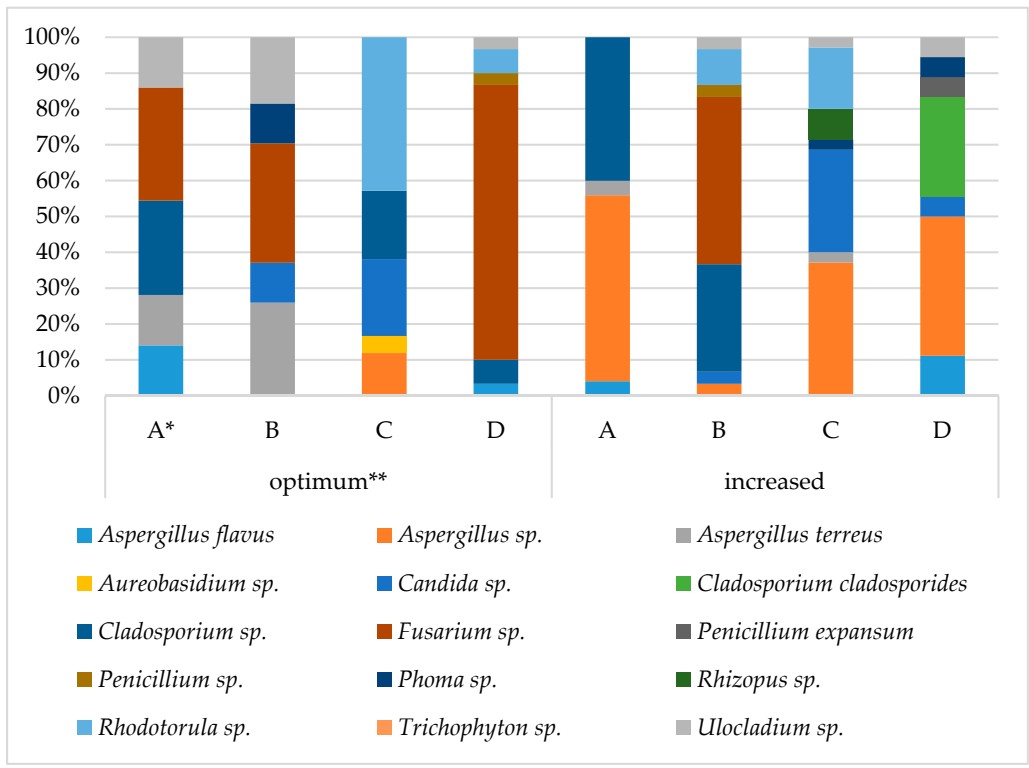

**Figure 5.** Identified mold fungi in the grain of spelt wheat cv. "Rokosz" (%) (mean for 2014–2015). Note: * crop protection methods, ** seeding density.

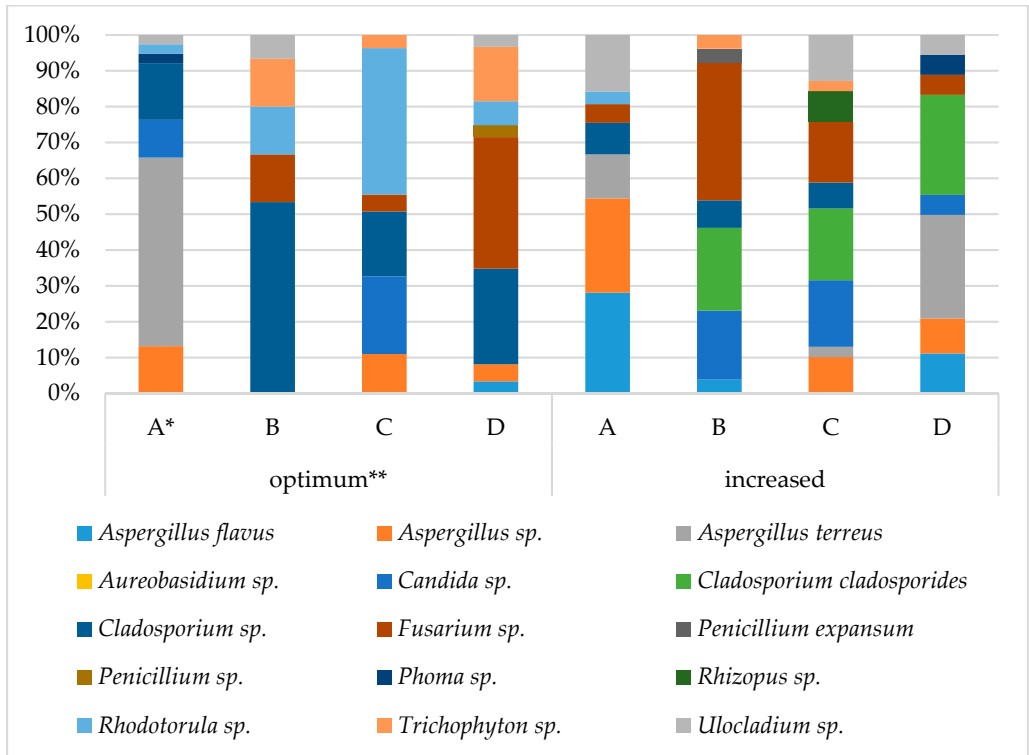

**Figure 6.** Identified mold fungi in grain of spelt wheat cv. "Schwabenspelz" (%) (mean for 2014–2015). Note: * crop protection methods, ** seeding density.

Both total mycotoxin content and the content of individual toxins in spelt grain were modified by the crop protection system and seeding density (Tables 8 and 9). In cv. "Rokosz" grain, significantly the highest number of mycotoxins was found in treatment C after application of herbicide crop protection alone, while the lowest number of mycotoxins was recorded in treatment D (after application of the fungicide) (Table 8). In grain harvested from the pro-ecological treatments (A and B), the number mycotoxins was at a similar level, but the amount of DAS, DON, ZEA, HT-2, and 15-acDON mycotoxins differed significantly. A denser seeding resulted in reduced grain contamination with mycotoxins. In cv. "Schwabenspelz", the lowest number of mycotoxins was found in the treatment with full chemical crop protection (treatment D) (Table 9). In the other treatments, on the other hand, grain contamination with mycotoxins was at a similar level and significantly higher than in treatment D. Similarly as in the case of cv. "Rokosz", a denser seeding contributed to a decrease in the number of mycotoxins in grain.

Regardless of the experimental factors, the amount of mycotoxins in grain of the spelt wheat cultivars studied did not differ significantly. One can only observe slightly greater susceptibility of cv. "Rokosz" to contamination with mycotoxins. In grain of this cultivar, the amounts of DAS and DON mycotoxins and of HT-2 toxin were found to be significantly greater than in "Schwabenspelz". Cv. "Rokosz" was found to have the highest content of NIV, 3-AcDON, and DAS mycotoxins, while cv. "Schwabenspelz" had the largest amounts of 3-AcDON, NIV, and 15-AcDON.

**Table 8.** Mycotoxin content in wheat grain of the spelt wheat cv. "Rokosz" (concentration of mycotoxins μg kg$^{-1}$ of grain) (mean for 2014–2015). NIV—nivalenol , DAS—diacetoxyscirpenol, 15- AcDON—15-acetyldeoxynivalenol, DON—deoxynivalenol, ZEA—zearalenone, 3-AcDON—3-acetyldeoxynivalenol.

| Experimental Factors | NIV | DAS | 15-AcDON | T-2 | DON | ZEA | HT-2 Toxin | 3-AcDON | Fuzarenon X | Sum |
|---|---|---|---|---|---|---|---|---|---|---|
| | | | | Crop protection | | | | | | |
| A | 243b | 167b | 78b | 13a | 203a | 6b | 5c | 100b | 51c | 866b |
| B | 277b | 216a | 145a | 11a | 89b | 51a | 49b | 47c | 24c | 909b |
| C | 317a | 69c | 126a | 20a | 75b | 7b | 154a | 299a | 205a | 1272a |
| D | 16c | 0.5d | 75b | - | 11c | - | - | 295a | 150b | 548c |
| | | | | Seeding density | | | | | | |
| optimum | 345a | 87b | 132a | 6a | 141a | 9a | 100a | 193a | 99a | 1111a |
| increased | 82b | 139a | 79b | 17a | 48b | 23a | 4b | 177a | 117a | 685b |

Different letters denote significant differences ($p \leq 0.05$). The same letter means not significantly different values.

**Table 9.** Mycotoxin content in wheat grain of the spelt wheat cv. "Schwabenspelz" (concentration of mycotoxins μg kg$^{-1}$ of grain) (mean for 2014–2015). NIV—nivalenol, DAS—diacetoxyscirpenol, 15- AcDON—15-acetyldeoxynivalenol, DON—deoxynivalenol, ZEA—zearalenone, 3-AcDON—3-acetyldeoxynivalenol.

| Experimental Factors | NIV | DAS | 15-AcDON | T-2 | DON | ZEA | HT-2 Toxin | 3-AcDON | Fuzarenon X | Sum |
|---|---|---|---|---|---|---|---|---|---|---|
| | | | | Crop protection | | | | | | |
| A | 227a | 47a | 135a | 31 | 6b | - | - | 259a | 132a | 836a |
| B | 245a | 61a | 61b | - | 54a | - | 27a | 176b | 90c | 712a |
| C | 221a | 60a | 110a | - | 14a | - | 11a | 206ab | 107bc | 727a |
| D | 39b | 5b | 110a | - | 11a | - | - | 165b | 128ab | 457b |
| | | | | Seeding density | | | | | | |
| optimum | 258a | 73a | 110a | 16 | 16a | - | 5a | 264a | 163a | 904a |
| increased | 108b | 13b | 98a | - | 26a | - | 14b | 139b | 65b | 462b |

Different letters denote significant differences ($p \leq 0.05$). The same letter means not significantly different values.

## 4. Discussion

Many authors indicate that the appropriate selection of a weed competitive variety is a major no-cost method of reducing weed infestation in crops [24,38–40]. In the opinion of Korres and Froud-Williams [41], as well as Mason and Spaner [42], high plants with intense tillering and forming a dense crop contribute to a reduced occurrence of weeds. This theory was confirmed by Kraska et al. [14], according to whom the selection of an appropriate cultivar had a significant impact on weed infestation of crops. Weeds produced significantly the lowest biomass in the tall spelt cultivar "Oberkulmer Rotkorn", while the highest biomass developed in the shortest spelt cultivar "Badengold". Although the research presented in this paper did not confirm statistically significant differences in the number of weeds found in the crops of the evaluated cultivars, at each time weed infestation was evaluated a significantly lower dry weight of weeds was determined in the tall cultivar "Schwabenspelz". Due to the higher competitive pressure of this cultivar, weeds were more shaded and had less space for growth, and hence they produced a lower biomass. Other research results confirm that spelt wheat is characterized

by high competitiveness against weeds, but they do not indicate unequivocally that this trait depends directly on plant height [24,43]. According to the study by Andruszczak et al. [40], the spelt cultivar "Frankenkorn", which was not significantly lower than many evaluated cultivars [43], was the most competitive against weeds, whereas according to Feledyn-Szewczyk [24] and Andruszczak et al. [39] the lowest weed infestation was found in the tall cultivar "Schwabenkorn" [43]. In the present study, the averages for the three-year study indicate that chemical crop protection (treatments C and D), based on its effect evaluated 3–4 weeks after application, significantly reduced the number and dry weight of weeds in the crops of both spelt wheat cultivars. The effect of the applied herbicide could be seen particularly in the first year of the study, in which the high rainfall in October caused intensive emergence of weeds. In 2014, the rainfall at the end of April and in May could have reduced the effectiveness of the herbicide. In the last year of the study, on the other hand, weed infestation of the crop was so low that the effect of herbicide action was not very visible. Herbicide application had no significant impact on weed infestation of cv. "Schwabenspelz" before harvest, but in the cv. "Rokosz" crop significantly lower weed infestation was shown in the treatments where herbicides were applied in comparison with the pro-ecological protection methods. In the studies by Andruszczak et al. [39,40], herbicide treatment significantly decreased the dry weight of weeds, but had no effect on their number. The results of the present study confirm the opinion of many authors who stress the low effectiveness of harrowing and its high dependence on weather conditions [44,45]. In the cv. "Rokosz" crop, weed infestation of the crop was not found to be significantly lower in the harrowing treatment compared to the control treatment in any year of the study. Nevertheless, in the first two growing seasons the number and biomass of weeds were slightly lower than in treatment A, whereas in 2015 weed infestation was slightly higher than in the harrowing treatment. In the cv. "Schwabenspelz" crop, harrowing also reduced weed infestation relative to the control treatment. According to Hansen et al. [46] and Lundkvist [47], the effectiveness of harrowing as a weed control method can be increased by performing this operation at an early time and repeating it two or three times, as well as by selecting weed-competitive crop plant cultivars. Despite the opinion of Korres and Froud-Williams [41] that an increase in seeding density of wheat contributes to reduced weed infestation of crops, this study did not demonstrate such a correlation for spelt wheat. It was attributable to the fact that a denser seeding did not cause a significant increase in ear density and therefore did not increase significantly the crop density.

The yield potential of crops is genetically driven and therefore selection of a variety with expected parameters is very important in crop production [14,22,32]. Moreover, in field cultivation a crop is affected by variable atmospheric conditions, which determine crop growth and development and, in consequence, yields obtained [16,22,28]. Spelt wheat is considered to be a species resistant to stress caused by adverse environmental conditions [25]. In the present study, the average grain yield for the period 2013–2015 was lower than those obtained by some researchers conducting studies in Poland [14,28]. This is due to adverse weather conditions in the first two years of the experiment, especially unevenly distributed rainfall in May and June, which contributed to the lower crop productivity. In a study by Wojtkowiak and Stępień [28], the average yield of cv. "Schwabenkorn" in 2013 was 7.47 t ha$^{-1}$, while a year earlier it was 46% lower. Jablonskytė-Raščė et al. [16] also found large differences in spelt yield in individual years of the study. This confirms the thesis that relatively small and unstable yields are a factor that limits the cultivation of spelt wheat [14]. Andruszczak [43] and Jablonskytė-Raščė et al. [16] found spelt wheat yields at a similar level compared to that obtained in the present study. In the study by Andruszczak [43], the yield of cv. "Schwabenspelz" was higher than in the present experiment, standing at 4.18 t ha$^{-1}$. According to a study conducted by Glamočlija et al. [21] in Serbia, on the other hand, yields of organically grown spelt wheat were lower, ranging from 2.33 to 3.35 t ha$^{-1}$. The study by Pospišil et al. [48,49], on the other hand, demonstrates that agronomic factors have a great impact on spelt wheat yield.

In many papers, it is stressed that spelt wheat is tolerant to different growing conditions and does not require intensive protection [9,21]. The results of conducted research indicate that the effect of crop

protection intensification on this species' yield is dependent on the variety. In the case of the Polish cultivar "Rokosz", a 19% higher yield was obtained after application of chemical crop protection (C, D) compared to pro-ecological cropping (A, B). Cv. "Schwabenspelz", on the other hand, did not respond with a substantial increase in yield after application of crop protection products, but the yields obtained were slightly higher than for the pro-ecologically grown crop. Andruszczak [43] demonstrated that crop protection intensification had a beneficial effect on the productivity of spelt wheat, whereas in the research carried out by Pospišil et al. [48] the impact of fungicide application on spelt wheat varied between years.

Similar to the studies conducted by Andruszczak [4] and Pospišil et al. [48,49], in the present experiment seeding density did not have a significant effect on most of the yield components and yield of spelt wheat. This finding is related to the phenomenon of the so-called crop self-regulation, due to which the number of plants per unit area is not directly proportional to the seeding rate. With increasing seeding rate, the phenomenon of self-regulation of the number of plants in a crop intensifies in order not to allow excessive density. At lower seeding densities, the tillering ability of the crop increases [12].

Wheat grains are colonized by various fungi, including plant pathogens and mycotoxin-producing species. The species composition of seed-colonizing fungi can be modified by various agronomic practices and weather conditions [23,50,51]. In this study, spelt grain was colonized by many fungi, but mainly by *Fusarium* sp., *Cladosporium* sp., and *Aspergillus* sp. The main mycotoxin producers are fungi of the genera *Aspergillus*, *Penicillium*, and *Fusarium* [52,53]. Species of the genus *Fusarium* differ in their epidemiology, pathogenicity, and ability to produce mycotoxins. Most *Fusarium* species can produce one or more mycotoxins exhibiting different degrees of toxicity [54,55]. Deoxynivalenol (DON), T-2, HT-2, and nivalenol (NIV) are the most common in cereals [56,57]. In the spelt grain evaluated, similar to products investigated by Serrano et al. [58], the mycotoxin NIV was found to have the highest content. In the opinion of Serrano et al. [58], special attention should be paid to this mycotoxin due to the toxicological effects of its occurrence and its large amount in cereal products. Spelt wheat is characterized by a higher level of resistance to fungal ear and grain infections than common wheat [8,59]. In grain of the spelt cultivars studied, all nine mycotoxins were found to occur, whereas Juan et al. [60] identified in spelt only four mycotoxins: DON, NIV, fuzarenon X (FUS-X), and enniatin A (EN A), whose amounts were comparable to the content of these compounds in the cultivar "Schwabenspelz". According to Mankevičienė et al. [23], lower DON, ZEA, and T-2/HT-2 concentrations were found in spelt grain without glumes than in spelt grain with glumes, in glumes, and in spring wheat grain. Likewise, Vučkovič et al. [8] detected higher levels of mycotoxins in hulls compared to grains. Conditions that are most favorable for the development of fungi and contamination of grain with mycotoxins include high humidity, frequent rainfall, particularly before harvest, and relatively warm night temperatures [61,62]. In the opinion of Blandino et al. [63,64], high plant density is associated with higher grain infection, while in the present study, grain of both cultivars contained fewer mycotoxins under conditions of denser seeding compared to less dense seeding. Such results could have been attributable to several factors. Firstly, the denser seeding did not cause an increase in ear density in the crop. Furthermore, despite that it was not confirmed statistically, under conditions of the higher seeding rate, the weed infestation of the crop before harvest of spelt wheat was slightly lower. Weeds occurring in the crop significantly affect crop health because they can change moisture conditions in the crop and host disease-causing fungi [65]. Lower infestation of spelt wheat resulted in lower fungal infection of grain and thus contributed to lower mycotoxin content. The effect of plant density on the amount of mycotoxins was investigated by many authors, but due to the diversity of factors affecting grain infection, their results are not unambiguous [66–68]. Mycotoxin content also depends on the cultivar [67,69]. In grain of both cultivars, in line with the expectations, the lowest amount of mycotoxins was found in the treatment with fungicide application. In cv. "Rokosz", most mycotoxins were found after application of the herbicide alone (treatment C). In cv. "Schwabenspelz", the amount of mycotoxins in grain obtained from treatments A, B, and C was

at a similar level. Grain of the evaluated spelt wheat cultivars, both in the pro-ecological treatments (A and B) and in the treatments with application of crop protection products, can be considered as safe and good because the content of individual mycotoxins did not exceed the maximum levels set for grain intended for food and animal feed purposes [70].

## 5. Conclusions

The present study showed how production intensification affected the productivity, weed infestation, and grain health of the spelt wheat cultivars "Rokosz" and "Schwabenspelz". The cultivar "Rokosz" was significantly less infested with weeds and produced significantly higher yields under chemical crop protection conditions than in the case of pro-ecological crop protection. The effect of crop protection methods on the quantitative weed infestation parameters and yield of this cultivar varied between years. Grain harvested from the treatment with herbicide application alone was characterized by the highest contamination with mycotoxins, while it was found to be lowest in the case of grain of grain harvested from treatment D. In the case of the cultivar "Schwabenspelz", no significant differences in yield were revealed between the pro-ecological and chemical crop protection treatments, and the effect of crop protection treatments on yield was not dependent on weather conditions. Cv. "Schwabenspelz" proved to be more competitive to weeds. At the dough stage of this crop plant, no significant differences in weed infestation were found between the pro-ecological treatments and those in which an herbicide was used. The lowest content of mycotoxins was determined in grain harvested from treatment D. In cv. "Schwabenspelz" grain, in comparison with the cultivar "Rokosz", the amount of mycotoxins DAS, DON, and HT-2 was determined be significantly lower, while the mycotoxin ZEA was not found at all.

Seeding density did not have a significant effect on most of the yield components, yield, and weed infestation of the spelt wheat crop. In both cultivars investigated, a higher seeding density contributed to reduced grain contamination with mycotoxins.

Due to the favorable response of the cultivar "Rokosz" to application of chemical crop protection, it is advisable to grow this species in conventional farms using integrated crop protection. The cultivar "Schwabenspelz", on the other hand, can be recommended for cultivation in organic farms on account of its competitive ability and resistance.

**Author Contributions:** The authors contributed to this article in the following ways—conceptualization: M.H., D.G., and B.N.-D.; data curation: D.G., S.C., and J.Ł.; formal analysis: M.H., D.G., B.N.-D., A.N., Ł.W., and M.R.-B.; funding acquisition: M.H.; investigation: M.H., D.G., B.N.-D., and A.N.; methodology: M.H., D.G., B.N.-D., A.N., and Ł.W.; project administration: M.H.; supervision: M.H. and D.G.; writing—original draft: M.H., D.G., B.N.-D., S.C., and J. Ł.; writing—review and editing: M.H., D.G., S.C., and M.R.-B. All authors have read and agreed to the published version of the manuscript.

**Funding:** Research supported by the Ministry of Science and Higher Education of Poland as the part of statutory activities of Department of Herbology and Plant Cultivation Techniques, University of Life Sciences in Lublin.

**Conflicts of Interest:** The authors declare no conflict of interest.

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
