# Peer review of "Evaluation of the Weed Infestation, Grain Health, and Productivity Parameters of Two Spelt Wheat Cultivars Depending on Crop Protection Intensification and Seeding Densities"

_agriculture, doi:10.3390/agriculture10060229_

Round 1
Reviewer 1 Report
The topic of the paper which aim at assessing the possibility of production intensification in spelt wheat is really interesting and worth to be published. A lot of work has been done to fulfill the objectives of the study and a lot of data are presented. Some improvements to the paper are needed before publication. I suggest writing in the material and methods which were the hypotheses of the study that brought to compare seeding density, different weed and pest control methods and different varieties. Moreover, as there are many tables and figures in the manuscript which make it really long I suggest choosing the most important information to communicate. Some tables presented non-significant data that can be cited in the text without adding a lot of tables.
In addition, it is not clear if in the ANOVAs the effect of crop protection methods, seeding density, crop variety, year and all the interactions have been tested for each considered variables. If some interactions were present it would be not correct to consider and analyze with post-doc tests each single factor as it has been done in the manuscript. Below some detailed comments to the manuscript.
Detailed Comments
Introduction
Well written, good number of citations of related works.
Material and Methods
Line 67: delete “is located” or rephrase the sentence
Table 1. please indicate in the caption if the value for each month is the average temperature.
Lines 75-80. As in the table are showed before rainfall amount and then temperature values, it could be better to describe them in the same order.
Lines 83-84: change “…were the object of this research” with “ …were evaluated in this study”
Line 87: change from “treatment without protection against pathogens” to “untreated control plots (no herbicides, fungicides and insecticide treatments)
Lines 85-103: explain what was the rationale of comparing 2 weed control techniques (harrowing, herbicides) and a full chemical application with herbicides, insecticide and fungicide. Why also applying insecticides if their effects were not tested? Then write a line to explain why you choose to compare an increased seeding rate. What did you expect? Which were the hypotheses?
Line 123: change to “ a sample of about 20 g of spelt grain….”
Also, indicate if a 20 g sample was taken from each plot.
Line 124: was homogenization done manually and for how many minutes? Indicate what was the dilution fluid.
Lines 136-138: was the method developed by the authors published? If yes, please add the citation
Line 143. Please indicate what was the “raw material”
Line 144: add “..were shaken with a centrifuge”
Line 162: change to: “Separation of trichothecenes, nivalenol and fuzarenone X was carried out…”
Lines 178-179: indicate on which variables (weed density, plant height…) the anovas were carried out.
Results
In the ANOVA analyses were the effects of crop protection method, seeding density, variety, year and their interactions tested? Which were the results? If there were interactions among factors you cannot consider the effect of single factor at a time as you have done. Please clarify. Also averaging data for the three-year study can only be done if there were no differences among years.
Line 203-204: delete the sentence “the same letter means not significantly different” in the table 2 and in all the other table captions as you already wrote that different letters means significantly different values.
In table 2 and following tables please indicate what “pcs” stands for.
In Tables 2, 3 and 4 the letters relative to the significant difference values were attributed in a different way compared to table 7 and 9 as in the first three tables the letter “a” was attributed to the significant highest values, while in tables 7 and 9 the same letter was attributed to the lowest values. Thus, it is necessary to be consistent throughout the tables for clarity.
In Table 2 and 4 the letters relative to the significant values need to be verified as they seem not to be attributed correctly in some cases. Check for example values of NW II 2013, the values 58.3 has really the letter “ac”?
Line 255-259. “In 2013 at the dough stage of wheat, seeding density caused significant differences only in the number of weeds per 1 m2 in the cv. ‘Schwabenspelz’ crop”. In Table 5 this difference is not significant. Where does this result come from?
Line 287 -caption of figure 1. Change to “The same letter means not significantly different VALUES”; please do the same at line 293 relative to Figure 2.
In figure 2 and 3 consider removing the letters on the graphs when there are no significantly different values as in the case of figure 1b or figure 2 b cv Rokosz as they are difficult to read. Where there were non-significant differences you can just write it in the figure caption or add ns on the graph (non-significant).
Line 270: change from “In the crops of both cultivars grown chemical crop protection…” to “ in both cultivars, chemical crop protection…”
Line 273-274: change from “plants was similar on all objects..” to “plants was similar among all the treatments..”
Lines 320-322: “In 2015 spelt wheat produced significantly longer ears in the treatments with an optimal seeding density relative to those obtained in the treatment with an increased seeding density”. This difference is not significant based on the letters of table 8.
Line 337-338: “treatment D was significantly higher…. in 2015 in comparison with treatments B and C.” in 2015 treatment D was not statistically different from C according to the letters indicated in table 9.
Line 345-346: “In 2013 the plant height in protection treatment D was significantly lower compared to that found in treatments B and C, whereas in 2015 in comparison with treatments A, B, and C”
According to the letters of table 9, in both 2013 and 2015 plant height in treatment D was significantly lower compared to all the other treatments. Please check the sentence and the table.
Lines 352-353: “On average over the three-year period, a higher grain yield of the spelt wheat cultivar ‘Schwabenspelz’ was obtained under higher seeding density conditions (Table 10)”. This is not visible in table 10 in which the small differences in yield appeared non-significant. Please check.
Table 11. Check the letters attributed to yield between the two cultivars as in this table the letter “a” was attributed to the lower values, while “b” to the higher values, but for yield it was done the opposite.
Lines 397-398: “In grain harvested from the pro-ecological treatments (A and B), the number mycotoxins was at a similar level”. This was not always true as in the case of DAS, DON, 15-acDON, ZEA, HT2…add that there were some exceptions.
Table 13. Change to cv. Schwabenspelz in the caption.
Table 12-13 and throughout the manuscript change the name of mycotoxins to the correct English names: i.e. nivalenol, Diacetoxyscirpenol, 15-Acetyldeoxynivalenol, Deoxynivalenol, 3-Acetyldeoxynivalenol…
Discussion
Lines 435-436: do the two varieties mentioned in these lines were spelt varieties? If yes, please indicate it.
Line 439: as the number of weeds between cultivars was similar but in the cv Schwabenspelz weed biomass was lower, this could be due to the fact that the weeds in this variety were smaller probably because of the competition and shading exerted by this taller variety. You could mention it in the discussion.
Lines 440-443: it is not clear why the citation of these related papers help in explaining your results. Was the most competitive variety also taller than the other? Please clarify.
Regarding the effect of the different crop protection methods on weeds, why in some cases the treatments C and D failed to reduce weed density and biomass compared to control and/or harrowing even at the first assessment? You should try to explain this result.
Lines 460-472: it is not necessary to list a long series of spelt wheat yields obtained in other studies, you can just mention some that agree with yours and some that found higher or lower yield. Thus, it would be better to shorten this list of yields and spend some more lines to discuss weed related results. For examples, harrowing was tested but in the discussion there is no mention of its efficacy compared to the other techniques.
Lines 508-511: some more lines should also be added to explain if the different crop protection techniques brought to higher or lower mycotoxin content. Did the absence of fungicide treatments enhance the mycotoxin content?
Lines 529-530: “In both cultivars under investigation, a higher seeding density contributed to reduced grain contamination with mycotoxins” do you have an explanation for this?
Author Response
Review (Round 1)
Answers to the comments of Reviewer 1
Thank you for your thoughtful, helpful, and in-depth review of our manuscript and for all your comments and suggestions. We hope that we have met the expectations and thus enhanced the quality of the manuscript. All the changes suggested by the Reviewer have been incorporated into the manuscript.
“Material and Methods”:
Line 67: Following the Reviewer’s comment, the words “is located” have been removed from the sentence.
Line 73: In Table 1 it is now indicated that these are mean temperatures.
Lines 75-80): Following the Reviewer’s comment, the description of precipitation has been placed before the description of air temperature, according to the order in which weather conditions are presented in Table 1.
Lines 83-84: Following the Reviewer’s comment, the words “…were evaluated in this study” has been put into the sentence instead of “…were the object of this research”.
Line 87: Following the Reviewer’s comment, the passage “treatment without protection against pathogens” has been changed to “untreated control plots (no herbicides, fungicides and insecticides treatments).
Lines 85-103: The research hypothesis has been added in the manuscript, which explains the justification for investigation of the factors included in the experiment.
Line 123: This part of the sentence has been changed to ”The sample of about 20 g of spelt grain…” and information has been added that samples were collected “from each plot”, in accordance with the Reviewer’s comment.
Line 124: Following the Reviewer’s comment, the following information has been added to the text regarding homogenization and dilution liquid: “Homogenized by hand for 5 min and then sedimented for 15 in sterile dilution liquid with approximately 100 microliters of Tween 80”.
Lines 136-138: The mycotoxin determination method described in the article has not been published in any paper yet.
Line 143: Information has been added that spelt grain was the raw material).
Line 144: Following the Reviewer’s comment, the passage “...were shaken with a centrifuge” has been added.
Line 162: The sentence “Separation of trichothecenes,nivalenol and fuzarenone X carried out” was changed to “Separation of trichothecenes, nivalenol and fuzarenone X was carried out…”.
Lines 178- Following the Reviewer’s suggestion, statistical method applied were analyzed thoroughly once again. After consultation with the statistician, it was concluded that the statistical analysis should be performed again (description in lines now 435-440). The results of the statistical analysis did not show significant relationships between crop protection system and seeding density for any of the characteristics studied. In the case of any trait where a significant interaction was found between the factor and year, interaction effects were analyzed separately for each year of the study. If no significant interactions were found between the factors, were compared differences between the means for main factors.
“Results”:
Following the Reviewer’s comment, the statistical analysis was verified and its description changed. Due to the repeated statistical analysis, the layout of the tables and their description have changed (chapters 3.1 and 3.2). Following the Reviewer’s suggestion, the number of tables and the amount of data given have also been reduced. In Table 2, weed infestation in the cv. ‘Rokosz’ depending on crop protection systems and years is analyzed, and pro-ecological (A, B) and chemical (C, D) crop protection is compared. Tables 3 and 5 have been replaced with graphs (Figure 1). Table 3 (previously Table 4) presents only the means for the study period. Tables 6, 11 and 14 have been removed and the data contained in them are only presented in descriptive form. Table 4 (previously Table 7) presents the means for the three-year study period. Table 5 shows cv. ‘Rokosz’ yields depending on crop protection and years. In Tables 6 and 7 (previously Table 9 and 10), only the means for the study period in accordance with the statistical analysis have been left. The based on the new statistical calculations, the letters relating to significant values were verified and changes have been made in the tables and in the description of the study results.
Lines 203-204: The sentence “the same letter means not significantly different” has been removed and now it has only been left in Table 2 and Figures 2, 3.
In Tables (now 2, 3, 4, 6, 7) and Figures (now 2, 3), the abbreviation “pcs” has been replaced with “no.” for better understanding of readers.
Following the Reviewer’s comment, in order to maintain the clarity, in all the tables the letters denoting significant differences between the values have been assigned in accordance with the following rule – letter “a” relates to the highest significant value.
After the repeated statistical analysis, the sentences inconsistent with the statistical analysis have been removed (the Reviewer’s comments in lines 255-259, 320-322, 337-338, 345-346, 352-353).
Line 287, 293: In the caption under Figure 2,3 the sentence “The same letter means not significantly different” has been changed to “The same letter means not significantly different values”.
In Figure 1b and 2b (now 2 b and 3b), we have not removed the letters denoting not significantly different values because we think that letter designations of even not significant differences are clearer.
Line 270: Following the Reviewer’s comment, the passage “In the crops of both cultivars grown chemical crop protection…” has been changed to “In both cultivars, chemical crop protection…”
Line 273-274: The passage “plants was similar on all objects…” has been changed to “plants was similar among all the treatments…”
Lines 397-398: Following the Reviewer’s comment, the description concerning mycotoxins has been improved.
Line 417: Following the Reviewer’s comment, in the caption of Table 13 (now Table 9), cv. ‘Rokosz’ has been changed to cv. ‘Schwabenspelz‘.
Line 411, 417: Following the Reviewer’s comment, in Table 12-13 (now Tables 8, 9) the names of mycotoxins have been changed to the correct English names.
“Discussion”:
Lines 435–436: It has been added that the two cultivars were spelt wheat cultivars.
Lines 439 and 440–443: Following the Reviewer’s suggestion, the discussion has been supplemented by adding a passage regarding the competitiveness of cv. ‘Schwabenspelz’ against weeds, citing the available literature concerning this issue.
In the chapter “Discussion” a passage has been added which explains the effect of different crop protection methods on weeds.
Lines 460–472: Following the Reviewer’s comment, the passage in the discussion regarding spelt wheat yields has been shortened, and the effectiveness of harrowing in weed control has been described.
Lines 508–511 and 529-530: More information has been added which explains which crop protection methods and seeding densities affect mycotoxin content.
Because new literature references have been added, the numbering of citations in the paper and in the “References” has changed.
The chapter "Conclusions" has changed partially due to the repeated statistical analysis.
Figures 5 and 6 have been re-inserted due to minor letters errors.
To facilitate the review, we applied a yellow color to all changes.

Reviewer 2 Report
The article is well-organized and contain all components: introduction, materials and methods, results, discussions, references. The introduction is well-developed and convinced me of the importance of spelt wheat varieties studies to determine how spelt wheat responds to production intensification. The article is well-written and easy to understand.
In order to indicate significant differences, the highest value was marked by the letter a (from Table 2 to Table 7). This should be done throughout the article.

Author Response
Review (Round 1)
Answers to the comments of Reviewer 2
Thank you for the review of our manuscript. We hope that we have met the expectations and thus enhanced the quality of the manuscript.
Following the Reviewer’s comment, in all the tables the letters denoting significant differences between the values have been assigned in accordance with the following rule – letter “a” relates to the highest significant value.
The following comments made regarding the manuscript have been incorporated into its revised version:
- In Table 13 (now Table 9) ’Rokosz’’ has been replaced with ‘Schwabenspelz’
- Jablonskytė-Rašč has been replaced with Jablonskytė-Raščė – line 461 and 467.
The chapter "Conclusions" has changed partially due to the repeated statistical analysis.
Figures 5 and 6 have been re-inserted due to minor letters errors.
To facilitate the review, we applied a yellow color to all changes.

Reviewer 3 Report
Dear authors,
your paper is very good ant the topic is interesting. The methodology of setting up experiments and implementation is consistent with other research. The results are very clear and useful to a wider (scientific and non- scientific) audience.
Generally everything is O.K. but the paper is too long with a lot of tables. If it is possible I suggest that you merge some tables, reduce the number of pages of results and increase the discussion page. You have ratio result:discussion 10:1,5.
These suggestions do not diminish my recommendation to accept the paper after minor corrections (in attach)
Best regards

Author Response
Review (Round 1)
Answers to the comments of Reviewer 3
Thank you for your thoughtful, helpful, and in-depth review of this manuscript and for all your comments and suggestions. We hope that we have met the expectations and thus enhanced the quality of the manuscript.
Following the Reviewer’s suggestions, the following changes have been made in the manuscript:
- Following the Reviewer’s suggestion, the number of tables and the description of the results have been reduced as well as the discussion has been expanded. Tables 6, 11 and 14 have been removed and the data contained in them are only presented in descriptive form.
- Line 20: Following the Reviewer’s comment , the passage “…to determine affect different crop” has been corrected to “to determine the effect of different crop”.
- Line 22: The word “present” has been removed.
- Line 26: “be can be” has been corrected to “can be”.
- Line 32: “Spelt wheat” has been put in Keywords instead of the Latin name “Triticum aestivum ssp. spelta”.
- In "Introduction" the sentences about the cropped area and yield of spelt wheat in Poland has been added.
- Line 54-55: Following the Reviewer’s comment, the passage “Studies conducted in different research centers…” has been changed to “Studies conducted by different researchers…”.
- The title of subchapter “2.1. Location of the Experiment and Soil and Climatic Conditions” has been changed to “2.1. Location of the Experiment, Soil and Climatic Conditions”.
- Line 67: The words “is located” have been removed.
- Line 69: The passage "...classified as good wheat soil complex and soil class II” has been removed since the soil classification system based soil complexed which is used in Poland is not an international classification.
- Line 72: The passage “…and a humus…” has been corrected to “…and a low humus…”.
- Lines 73-74: Following the Reviewer’s comment (), in Table 1 the data regarding the long-term average have been added.
- Lines 75-80: The years of the study are now compared with the long-term average.
- Lines 93-95: The repeated names of active substances have been removed.
- Lines 100-103: We could not harmonize for both cultivars the used names of seeding material - “seeds” or “spikelets”. The Polish cv. ‘Rokosz’ is threshable and in the case of this cultivar kernels are used as seeding material. But the threshability of cv. ‘Schwabenspelz’ is very poor, after threshing seeds are often damaged, and due to this their germination ability is lower. For this reason, in the case of this cultivar spikelets are used as seeding material.
- Line 107: The sentence “The sown and harvested plot area was 13.5 m2.” has been changed to “The sown and harvested plot area was 13.5 m2, per repetition”.
- Line 112: The word “Approximately…” has been added in the sentence.
- Lines 123-132: Following the Reviewer’s suggestion, in chapter 2.4. the methodology has been shortened.
- Lines 143-160: But it is not possible to shorten the procedure for preparing extracts and the course of chromatographic analysis due to readability and possible reconstitution of the experimental part by potential readers who want to carry out tests according to the proposed solution. The methodology was not published in any journal.
- Line 188: Due to the change in the number of tables and the removal of a part of their description, it was not necessary to change the word “exhibited” to “showed”.
- Following the Reviewer’s suggestion, in lines 198, 216, the headings in the tables have been shortened, leaving only the data necessary, in our opinion, for the readability of the results.
- line 230: In the commentary to the table, years of research were not added because in Table 3 (previously Table 4), only the means for the study period are presented.
- Line 270: The words “the crops of” and “grown” have been removed from the sentence (now 322).
- Line 273: (BBCH -83-85) has been corrected to (BBCH 83-85).
- Line 316: In this sentence, “modified the” has been changed to “modified only the”.
- Lines 378:Following the Reviewer’s suggestion have been added „statistically significantly lower (p≤0.05)”
- Lines 381-382: Following the Reviewer’s suggestion, the sentence has been changed to “„The fungi Penicillium sp., Penicillium expansum, Candida sp., Aureobasidium sp., Aspergillus sp., Aspergillus flavus, Aspergillus terreus, Cladosporium sp., Cladosporium cladosporides, Ulocladium sp., Fusarium sp., Rhodotorula sp., Rhizopus sp., Trichophyton sp., Phoma sp. were found in diffrent ratio in spelt wheat (Figures 4 and 5).”
- Lines 412: The analysis of mycotoxin content was performed on grain harvested during two growing seasons and therefore we could present only the results for the years 2014-2015.
- Following the Reviewer’s suggestion, (lines 429) the discussion chapter has been expanded
- Lines 457: The expression “by many” has been replaced with “by some”.
- Lines 513: Following the Reviewer’s suggestion, this sentence “The present study showed production intensification to have an impact on the productivity, weed infestation, and grain health of the spelt wheat cultivars ‘Rokosz’ and ‘Schwabenspelz’ has been corrected to “The present study showed how production intensification has impact on the productivity, weed infestation, and grain health of the spelt wheat cultivars ‘Rokosz’ and ‘Schwabenspelz’”.
- Line 525-526: The inappropriately formulated sentence has been corrected. The sentence “Despite that no significant differences in grain contamination with mycotoxins were revealed between the cultivars in question, the cultivar ‘Schwabenspelz’, in whose case contamination was lower by 24%, can be considered to be more resistant” has been changed to “In cv. ‘Schwabenspelz’ grain, in comparison with the cultivar ‘Rokosz’, the amount of mycotoxins DAS, DON, and HT-2 was determined be signficantly lower, while the mycotixin ZEA was not found at all.”
- Because new literature references have been added, the numbering of citations in the paper and in the “References” has changed.
- The chapter "Conclusions" has changed partially due to the repeated statistical analysis.
- Figures 5 and 6 have been re-inserted due to minor letters errors.
- To facilitate the review, we applied a yellow color to all changes.

Round 2
Reviewer 1 Report
The authors made a lot of efforts in answering the questions posed by the reviewers. I also appreciated that the statistical analyses have been re-run.
The manuscript has deeply improved after revision and it can now be published with some minor changes:
Table 2. Please check again the letters of the significance. Why in 2015 the comparison among control methods in NWI are all “c”? if there is no significance they should be all “a” or better just put ns (non-significant). the same in 2015 WWI and 2014 WWI and WWII.
Line 213. Change: “where the herbicide was applied” with “where the herbicides were applied”.
Table 5. Also here check the letters of the significance for all years. The letters have not been correctly attributed to the values. i.e. Why in 2014 there is no “a”? or in 2015 there is “ab” but not “b”.
Line 343: correct the spelling of the word “different”
Line 395: delete “occurring in its crop” to read “…weeds were more shaded…”
Line 402: delete “the crop” and change “high cultivar” with “tall cultivar” to read “ …the lowest weed infestation was found in the tall cultivar..”
Line 411: change “where the herbicide was applied in comparison” with “where herbicides were applied in comparison ”
Line 483: check the citations, there is a citation with an author’s surname left (Blandino).
Author Response
Replies to the comments
Thank you very much for your correct and comments. Regarding to lines 213, 343, 395, 402, 411 and 483 all mistakes has been corrected. By introducing letter designations, the authors wanted to obtain an effect indicating that, for example, average NWI values for a combination of factors:
- (year =”2013”; Crop protection = „A”), (year = „2013”; Crop protection = „B”) and (year = „2014”; Crop protection = „A”) do not differ significantly (the letter "a")
- (year = „2013”; Crop protection = „B”), (year = „2014”; Crop protection = „A”), (year = „2014”; Crop protection = „B”) and (year = „2014”; Crop protection = „D”) they not differ significantly (the letter "b")
- The same rule applies to the following features
- The same applies to the features in Table 5.
Best regards
